# Recovery of Tropical Cyclone Induced SST Cooling Observed by Satellite in the Northwestern Pacific Ocean

**Zheng Ling** [1,2,3], **Zhifeng Chen** [1], **Guihua Wang** [3,4], **Hailun He** [2] and **Changlin Chen** [4,*]

1 Laboratory of Coastal Ocean Variation and Disaster Prediction, College of Ocean and Meteorology, Guangdong Ocean University, Zhanjiang 524088, China; lingz@gdou.edu.cn (Z.L.); wang5@stu.gdou.edu.cn (Z.C.)
2 State Key Laboratory of Satellite Ocean Environment Dynamics, Second Institute of Oceanography, Ministry of Natural Resources, Hangzhou 310012, China; hehailun@sio.org.cn
3 Southern Marine Science and Engineering Guangdong Laboratory (Zhuhai), Zhuhai 519080, China; wanggh@fudan.edu.cn
4 Department of Atmospheric and Oceanic Sciences & Institute of Atmospheric Sciences, Fudan University, Shanghai 200433, China
* Correspondence: chencl@fudan.edu.cn

**Abstract:** Based on the satellite observed sea surface temperature (SST), the recovery of SST cooling induced by the tropical cyclones (TCs) over the northwestern Pacific Ocean is investigated. The results show that the passage of a TC induces a mean maximum cooling in the SST of roughly $-1.25\,^{\circ}$C. It was also found that most of this cooling (~87%) is typically erased within 30 days of TC passage. This recovery time depends upon the degree of cooling, with stronger (weaker) SST cooling corresponding to longer (shorter) recovery time. Further analyses show that the mixed layer depth (MLD) and the upper layer thermocline temperature gradient (UTTG) also play an important role in the SST response to TCs. The maximum cooling increases ~0.1 $^{\circ}$C for every 7 m decrease in the MLD or every 0.04 $^{\circ}$C/m increase in the UTTG. The combined effects of MLD and TC intensity and translation speed on the SST response are also discussed.

**Keywords:** sea surface temperature; tropical cyclones; mixed layer depth; thermocline temperature gradient

## 1. Introduction

Gaining insight into the tropical cyclone (TC) induced SST cooling is important for both improving TC prediction and understanding the atmospheric and oceanic circulation, which not only influences the number, path, and intensity of the TCs [1–7] but also affects the large-scale atmospheric circulation [6] and oceanic thermohaline circulation [8–10]. TC-induced SST cooling has been widely studied in recent decades [11–19]. The maximum SST cooling generally occurs to the right of the TC track in the northern hemisphere [15,20–22], although it can be observed to the left of the TC track [23,24] or exactly along the track [25]. The magnitude of SST cooling induced by the passage of TCs has been reported to range from 1 $^{\circ}$C to 10 $^{\circ}$C [14,26–29]. Dare and McBride [30], using a compositing approach, studied the response of SST to the passage of TCs in the global ocean and found that the maximum cooling occurs ~1 day after TC passage and mean maximum SST cooling is about $-0.9\,^{\circ}$C.

After the passage of a TC, there is a recovery period reported in previous studies to range from several days to more than a month. For example, Hazelworth [31] found that the recovery time ranged from 1 to 36 days with an average duration of about 20 days. However, Nelson [32] noted that SST cooling induced by Hurricane Felix had not disappeared nearly one month after the TC passage. The result of Hart et al. [6] from 1985 to 2002 showed that the average time for the SST to return to climatological values is about 35~40 days in the Northern Hemisphere. Globally, roughly 44% of the SST cooling below

the climatological values had disappeared within 5 days of storm passage, and roughly 88% of it had disappeared prior to 30 days post-storm [30].

The magnitude of SST cooling is influenced by both the surrounding ocean environment and the characteristics of the TC. TC intensity (TCI) [6,14,27,33] and TC translation speed (TCTS) [14,27,34] will affect the cooling; stronger, slower moving, and/or larger spatial scale TCs cause larger SST cooling. In addition, the thermal structure of the upper ocean is also important in determining the overall SST cooling response to the passage of a TC. The thermal structure of the upper ocean is largely specified by the mixed layer depth (MLD) and upper thermocline temperature gradient (UTTG) [14], with stronger SST cooling being associated with shallower MLDs and sharper UTTGs. TCs cause SST cooling through several dynamic processes, such as entrainment at the bottom of the mixed layer, air-sea heat exchange, and upwelling. Among these processes, entrainment appears to play the primary role in SST cooling [14,25,35,36].

While most studies of TC-induced cooling focused on the passage of individual storms, Dare and McBride [30] studied the SST response to the passage of TCs over the entire globe. However, they only considered the effects of the TCI and TCTS and did not include information on the upper ocean's thermal structure. Previous studies have shown that the ocean thermal conditions also have an important impact on the interaction between a TC and the ocean [37–40]. In the present paper, we will focus on the effects of the oceanic thermal structure (including MLD and UTTG) on the SST response to the passage of TCs in the northwest Pacific (NWP) Ocean, the most active ocean basin with respect to TCs. In addition, the combined effects of the characteristics of the TCs (TCI and TCTS) and the ocean (MLD and UTTG) are also discussed.

## 2. Data and Method

### 2.1. Data

The TC dataset named 'International Best Track Archive for Climate Stewardship (IBTrACS)' is from National Oceanic and Atmospheric Administration (NOAA, https://www.ncei.noaa.gov/data/international-best-track-archive-for-climate-stewarship-ibtracs/v04r00/access, accessed on 1 November 2019) [41]. The dataset contains TC data from 14 institutions and the data from the Chinese Meteorological Administration (CMA) was used in the present study as justified by previous studies [42,43]. This data contains three-hourly TC positions (most data reported at 6 h intervals but interpolated to 3 h by IBTrACS) and two-minute mean maximum sustained wind (MSW) near the TC center. The TCI is divided into four storm categories based on the MSW: tropical depression (TD, MSW under 33 kn), tropical storm (TS, MSW between 34 kn and 47 kn), severe tropical storm (STS, MSW between 48 kn and 63 kn), and typhoon (TY, MSW above 64 kn) [44]. The dataset period used in the present paper is from 1999 to 2018 in which 555 TCs are detected and tracked.

The satellite observed SST used in the present paper is the multi-sensor L4 foundation SST product from Remote Sensing Systems (REMSS, www.remss.com, accessed on 12 April 2019), which is designed to represent a foundation SST at a depth of ~1 m or temperatures just below the diurnal layer. This dataset is observed by satellite-based microwave radiometers, which can measure the SST under cloud cover during the typhoon period, owing to the weak absorption of microwaves by cloud droplets. The temporal and spatial resolutions of these SST datasets are daily and 0.25°, respectively.

The monthly ocean temperature dataset is the ocean gridded product on a $1° \times 1°$ grid [45] from the Institute of Atmospheric Physics (IAP, http://159.226.119.60/cheng/, accessed on 14 September 2019), which is used to calculate the MLD and the temperature gradient at MLD (used to represent the UTTG). Another supplementary ocean temperature dataset consists of the Argo temperature profiles, which are from the China Real-time Argo Data Center (ftp://ftp.argo.org.cn/pub/ARGO/global/, accessed on 29 August 2020). The above two datasets cover the period from 1999 to 2018 are used in the present paper. The Argo temperature profiles are used to calibrate the MLD and UTTG calculated from the IAP dataset.

### 2.2. Method

The TC-induced SST cooling was calculated using the following three steps: First, the 0.25° × 0.25° gridded SST data were linearly interpolated to estimate the SST at the location of the center of each TC at the observation times. The linear interpolations were repeated for every observed TC position along the TC storm track for the period from 7 days before TC passage to 60 days after TC passage. As a result, a time series of over 18,000 SST values for each observed position of a TC center was obtained. Second, in order to remove the influence of the seasonal variability of SST, the daily climatological SST for the same day and location was subtracted from each point in the time series of SST to get the time series of SST anomalies (SSTAs) [30]. The daily climatological SST was constructed by averaging values at each TC location from the 1999–2018 daily SST dataset. Third, the initial SSTA conditions 3 days prior to TC passage were subtracted from each SSTAs time series [9] to obtain an estimate of the magnitude of TC-induced SST cooling. The recovery time was then defined as the time it takes for the TC-induced SST anomaly to disappear, indicating a return to the pre-TC SST value (SSTA on D−3, D−3 means 3 days before TC passage) for every location over which a TC passed.

To calibrate the MLD and UTTG calculated from the IAP monthly temperature dataset, we first need to select the corresponding Argo temperature profiles. The choice of the Argo temperature profiles is based on the following two criteria: The first one is that the observed time of the Argo temperature profiles must be within 3–10 days prior to the TC passage. The second is that the distance between the Argo floats and the TC centers must be less than 50 km and if more than one float satisfies this criterion, the one with the shortest distance is used. Using these criteria 891 Argo temperature profiles were obtained.

Figure 1 shows the MLD and UTTG calculated from the Argo temperature profiles and the IAP dataset. The MLD is defined as the depth where the temperature is 0.2 °C different from 10 m temperature [46] and the UTTG is the temperature gradient at the bottom of the mixed layer [14]. The correlation coefficients of MLD and UTTG between the Argo and IAP datasets are 0.65 and 0.62, respectively. Both coefficients are significant at the 99% confidence level, based on Student's t-test. Using least-squares linear regression, we obtained the following two regression functions:

$$\text{MLD}_{\text{-Argo}} = 1.09 \times \text{MLD}_{\text{-IAP}} \tag{1}$$

$$\text{UTTG}_{\text{-Argo}} = 1.39 \times \text{UTTG}_{\text{-IAP}} \tag{2}$$

The R-squared values of these two fitting functions are 0.85 and 0.61, respectively. Due to the spatial-temporal smoothing, the magnitudes of MLD and UTTG in the IAP dataset are smaller than those in the Argo dataset, this effect is especially pronounced for the UTTG magnitudes. Generally, the Argo dataset is more convincing. However, the Argo dataset has fewer data points (891), which will allow outliers to have a greater impact on the overall result and potentially cause the result to lose statistical significance. As mentioned above, from a display of the correlation coefficients between the two datasets, the MLD and UTTG of the IAP dataset have a good match with those of the Argo dataset. Therefore, because of its more extensive coverage and to allow for a more detailed analysis, the IAP dataset is chosen for the following analyses and the Argo dataset is used to calibrate the IAP dataset. Based on the fitted regression functions (1) and (2), the magnitude of MLD and UTTG computed from the IAP dataset is amplified by the multiplicative scale factors of 1.09 and 1.39, respectively for use in our analysis.

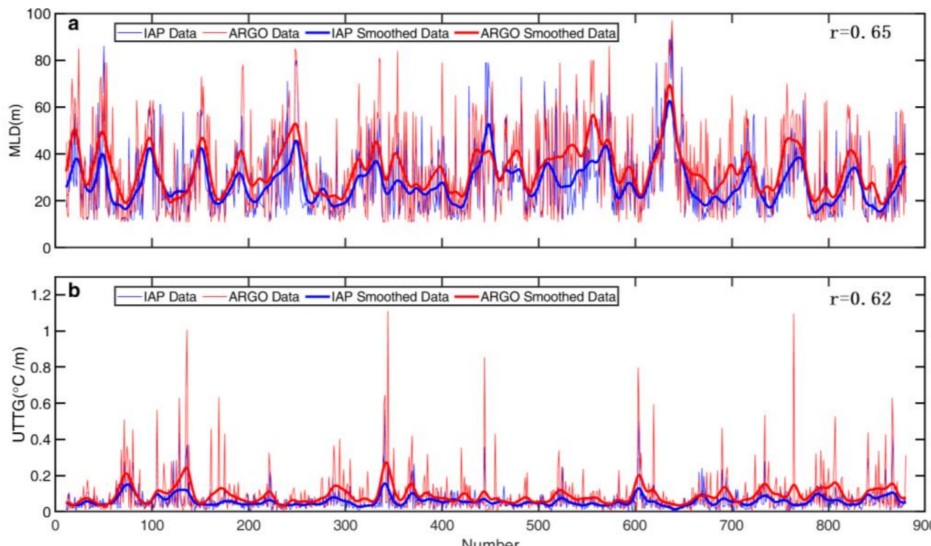

**Figure 1.** Comparison between (**a**) the MLD (m) and (**b**) UTTG (°C/m) calculated from the IAP dataset (blue lines) and the Argo dataset (red lines). The thin and weighted lines represent the original and 21-point smoothed data, respectively.

## 3. Results

### 3.1. Characteristics of SST Response to TC Passage

The time of maximum cooling relative to TC passage was compiled from 1 day before to 7 days after TC passage (D−1 to D7) and the result is shown in Figure 2. This result shows that the maximum SST cooling occurs most frequently on D1 after passage (Figure 2a), with the corresponding mean maximum cooling on each day ranging from −0.58 °C on D−1 to −1.47 °C on D1, with an overall mean value of −1.25 °C (Figure 2b). These results are consistent with those of Dare and McBride [30], except that the magnitude of cooling found here (−1.25 °C) is significantly larger than theirs (−0.9 °C). The reason may be that the SSTA they used was relative to the climatological value while the value we used is relative to the initial SSTA conditions (SSTA on D−3). This difference suggests that the interannual variability of SST cannot be neglected when estimating TC-induced cooling of the SST.

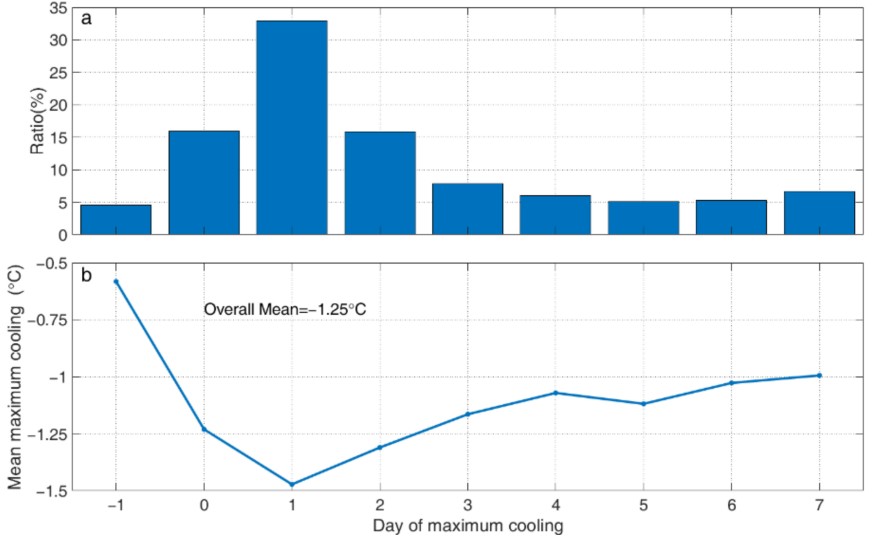

**Figure 2.** (**a**) Observed frequency (%) of the day of maximum cooling occurrence and (**b**) the corresponding mean maximum cooling for each day. D0 denotes the time of TC passage.

Figure 3 shows the frequency of the maximum cooling in 0.5 °C bins and the corresponding SST recovery time. As shown in Figure 3a, the majority of the TC passages (64%) induce maximum TC-induced SST cooling of less than 1.5 °C with the two highest frequencies occurring in −1 °C and −0.5 °C bins. As expected, the recovery time is directly related to the magnitude of the maximum cooling, which increases with the stronger maximum cooling (Figure 3b). However, it should be noted that the variation in the recovery time in all but the 0 °C bin is quite large, with the standard deviations being 13.6 and 6.8 days for −1 and −0.5 bins, respectively, and larger than 17 days for the other bins.

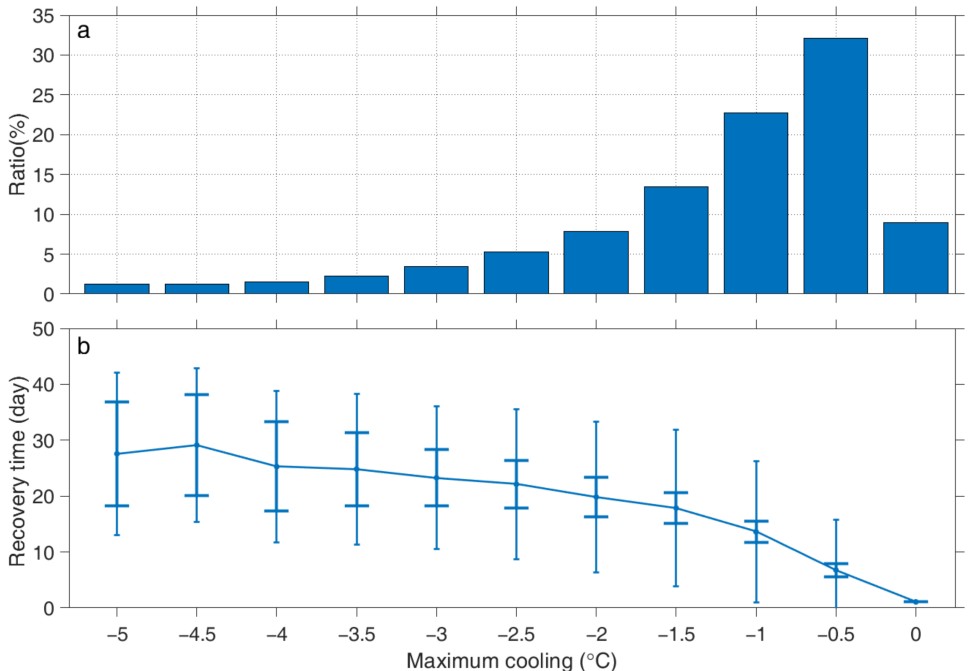

**Figure 3.** (**a**) Observed frequency (%) of the TC-induced SST cooling occurrence in each 0.5 °C bin and (**b**) the corresponding recovery time for each bin. The thin and bold error bars represent the standard deviation of the individual value and the mean, respectively. The standard deviation of the mean is multiplied by 10 in the plot.

The time series of mean TC-induced SST cooling composited into recovery time ranges of 1–5 days, 6–30 days, and larger than 30 days are shown in Figure 4. The relative frequencies of the three composite time series bins are 39.8%, 47.6%, and 12.6% for the 1–5 days, 6–30 days, and larger than 30 days recovery time bins, respectively. Generally, the longer recovery times are associated with stronger maximum TC-induced SST cooling (Figure 4a). For the mean time series for the three composite bins, the maximum TC-induced SST coolings are −0.34 °C, −1.32 °C, and −1.79 °C, respectively and the maximum cooling occurrence time is D1 for all bins. For the mean time series relative to the day of maximum TC-induced SST cooling (Figure 4b), the maximum TC-induced SST coolings are −0.56 °C, −1.56 °C, and −2.04 °C for the three bins, respectively. In general, following the day of maximum cooling, there is a rapid recovery period of about 2 days duration (average value is about 0.49 °C), which is followed by a slow recovery period (Figure 4b).

As mentioned in the introduction, the magnitude of TC-induced SST cooling is affected by several factors, including TCI, TCTS, and the MLD and UTTG in the ocean. Since Dare and McBride [30] have discussed the impacts of the intensity and translation speed of the TCs, this paper will focus on the effects of the oceanic thermal parameters on SST response to the TCs and will consider the combined effects of the oceanic and TC's factors on the SST response to the passage of TCs.

### 3.2. Impacts of the Ocean MLD and UTTG

To study the effect of MLD on the SST response to the passage of a TC, we divided the observed SST responses into five 20 m wide bins based on pre-passage MLD (Table 1). The 0–20 m, 20–40 m, 40–60 m, 60–80 m, and >80 m bins account for 30.8%, 39.1%, 21.8%, 6.1%, and 2.2% of the observations, respectively. As shown in Figure 5a, increases in the MLD cause the maximum cooling to decrease throughout the cooling period and the time of the strongest maximum cooling to occur later. The reason for this is that the cold water under MLD is more difficult to entrain and needs more time to be brought to the sea surface by mixing for a deeper MLD. The difference of maximum cooling between D1 to D7 is relatively small, particularly for a shallower MLD. The fact that the maximum cooling decreases with the increase of the MLD is clear in Figure 5b, in which the cooling decreases from −1.62 °C for MLDs shallower than 20 m to −0.63 °C for MLDs greater than 80 m. As mentioned above, the recovery time has a strong positive correlation with the maximum cooling, also decreasing gradually from 14.4 days for MLDs shallower than 20 m to 10.2 days at MLDs deeper than 80 m.

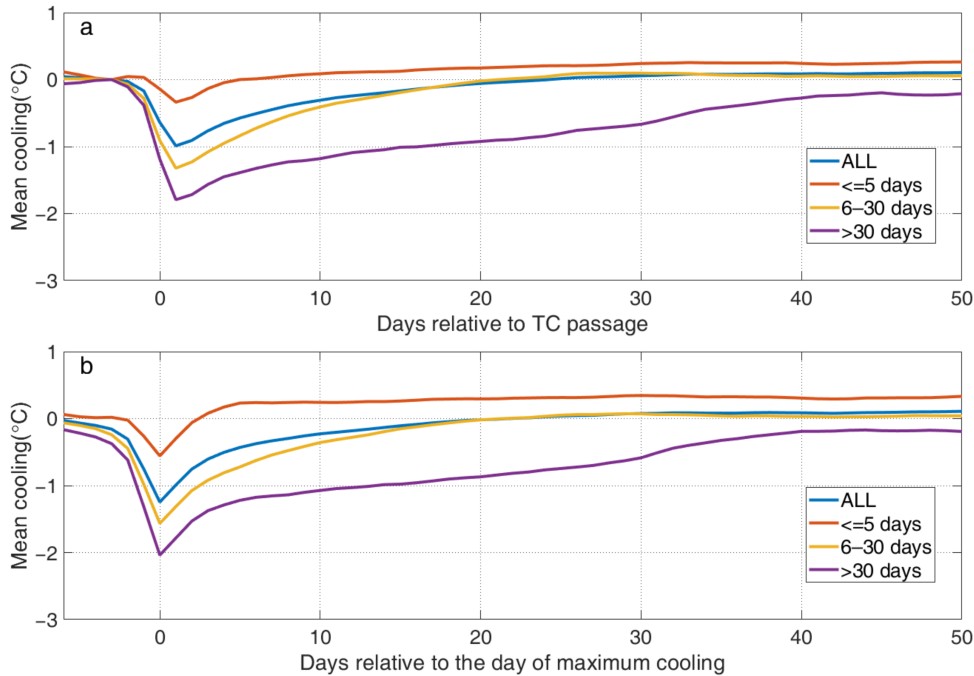

**Figure 4.** The composite time series of TC-induced SST cooling relative to (**a**) the day of TC passage and (**b**) the day of maximum cooling based on different recovery time.

**Table 1.** The average and standard deviation of maximum cooling and recovery time and the observational frequency distributions (%) for different MLD bins.

| Bin | MLD (m) | Mean Maximum Cooling (°C) | Mean Recovery Time (Days) | Observations (%) |
|-----|---------|---------------------------|---------------------------|------------------|
| 1 | 0–20 | −1.62 ± 1.20 | 14.4 ± 13.9 | 30.8 |
| 2 | 20–40 | −1.24 ± 1.03 | 13.4 ± 13.6 | 39.1 |
| 3 | 40–60 | −0.92 ± 0.87 | 11.1 ± 12.6 | 21.8 |
| 4 | 60–80 | −0.76 ± 0.72 | 9.8 ± 11.9 | 6.1 |
| 5 | >80 | −0.63 ± 0.43 | 10.2 ± 12.5 | 2.2 |

As mentioned by Price [14], the UTTG is another factor that is important in determining the SST response to the passage of TCs. As was done for the MLD, we also divided the SST responses into five bins based upon the pre-passage UTTG (Table 2), with the five bins listed in Table 1 accounting for 29.0%, 25.4%, 16.3%, 11.0%, and 18.4% of the observations, respectively. The tabulated results along with those shown in Figure 6b

demonstrate that the maximum cooling increases as the UTTG increases. Comparing the UTTG dependence of the cooling, ranging from −1.03 °C to −1.55 °C, to that due to a changing MLD, which ranges from −0.63 °C to −1.62 °C, indicates that the effect of UTTG on the maximum cooling is weaker than that of MLD (Figure 6a). The magnitudes of mean maximum cooling in the five UTTG bins are −1.03 °C, −1.20 °C, −1.27 °C −1.38 °C, and −1.55 °C, with associated recovery times of 12.1, 12.9, 13.5, 12.9, 13.9 days (Table 2), respectively. It seems that the UTTG does not play as strong a role in the recovery time as the MLD, as the difference in the recovery time among the five UTTG bins is only 1.8 days, while that for the MLD bins is 4.6 days. It should be noted that the standard deviations of maximum cooling and recovery time are large for all MLD and UTTG bins (Tables 1 and 2), indicating that other factors (such as TCI, TCTS) also play an important role in the SST response to the TCs.

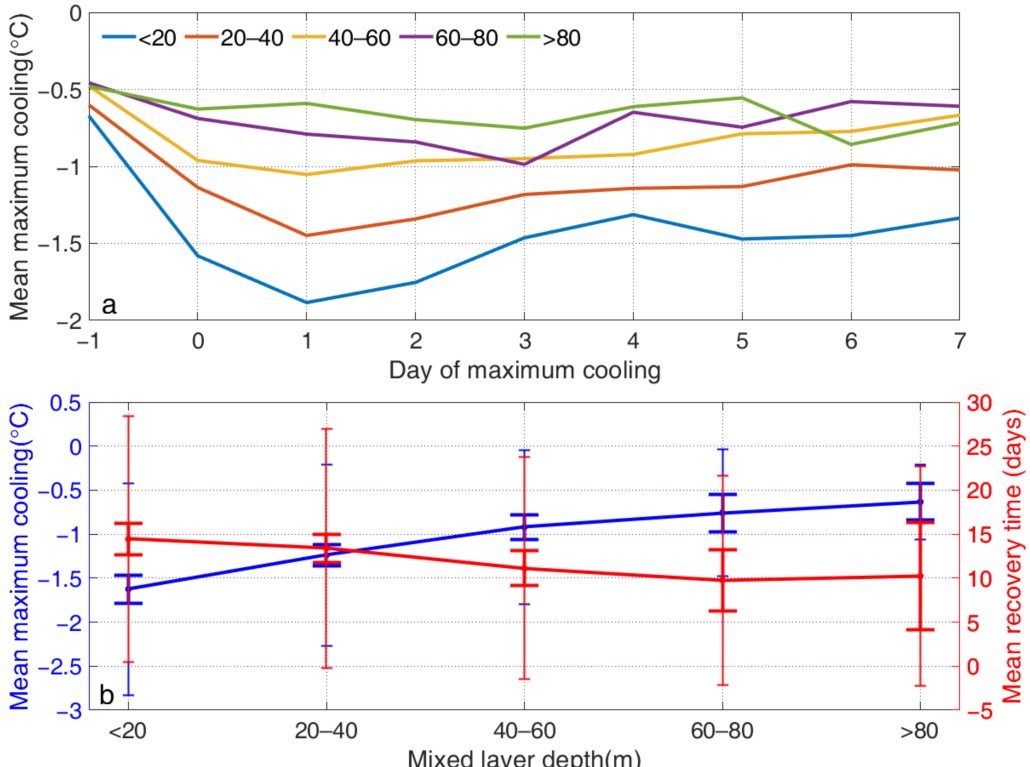

**Figure 5.** (**a**) Mean maximum cooling (°C) on each day in different MLD bins and (**b**) the average and standard deviation of maximum cooling (°C) and recovery time (days) in different MLD bins. D0 denotes the day of TC passage. The thin and bold error bars represent the standard deviation of the individual value and the mean, respectively. The standard deviation of the mean is multiplied by 10 in the plot.

**Table 2.** The average and standard deviation of maximum cooling and recovery time and observational frequency distributions (%) for different UTTG bins.

| Bin | UTTG ($10^{-2}$ °C/m) | Mean Maximum Cooling (°C) | Mean Recovery Time (Day) | Observations (%) |
|---|---|---|---|---|
| 1 | 0–4 | −1.03 ± 0.91 | 12.1 ± 12.7 | 29.0 |
| 2 | 4–6 | −1.20 ± 1.01 | 12.9 ± 13.5 | 25.4 |
| 3 | 6–8 | −1.27 ± 1.10 | 13.5 ± 14.2 | 16.3 |
| 4 | 8–10 | −1.38 ± 1.18 | 12.9 ± 13.2 | 11.0 |
| 5 | >10 | −1.55 ± 1.20 | 13.9 ± 13.9 | 18.4 |

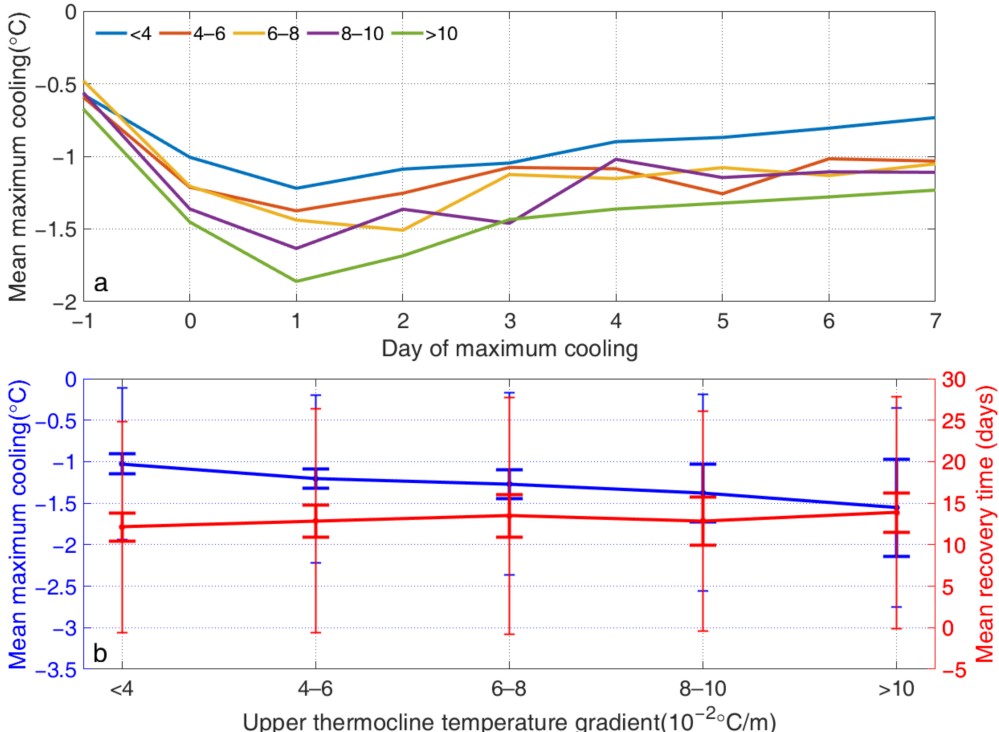

**Figure 6.** (**a**) Mean maximum cooling (°C) on each day in each UTTG bin and (**b**) the average and standard deviation of maximum cooling (°C) and recovery time (days) in each UTTG bin. The thin and bold error bars represent the standard deviation of the individual value and the mean, respectively. The standard deviation of the mean is multiplied by 10 in the plot. D0 in (**a**) denotes the day of TC passage.

As mentioned in the introduction, in addition to oceanic factors, the SST response to the passage of TCs is also affected by the TCI and TCTS. As the direct effects of these latter factors have been discussed elsewhere [30], we will focus on the effects of the interaction between oceanic thermal structure and the properties of TCs on SST cooling. As it appears to be most important, the primary focus will be on the interaction between the TC properties and the MLD.

As shown in Figure 7a, the maximum cooling increases with increasing TCI for almost every MLD bin, except for the deepest MLD bin of TS. However, this result may have too few data points (419 data points) for the MLDs greater than 80 m (Table 1) to be a robust result. In addition, the general variation in the maximum cooling and recovery time for different TCIs in each MLD bin are the same: the maximum cooling and recovery time both decrease with increasing MLD. However, it should be noted that the effect of MLD weakens as the TCI decreases. As shown in Table 3, for TD, the difference in maximum cooling between two adjacent MLD bins is smaller, with an average difference of only about 0.10 °C. However, for TY, the average difference between two adjacent MLD bins is about 0.37 °C, suggesting that the effect of MLD on the maximum cooling is greater for intense storms than weak ones.

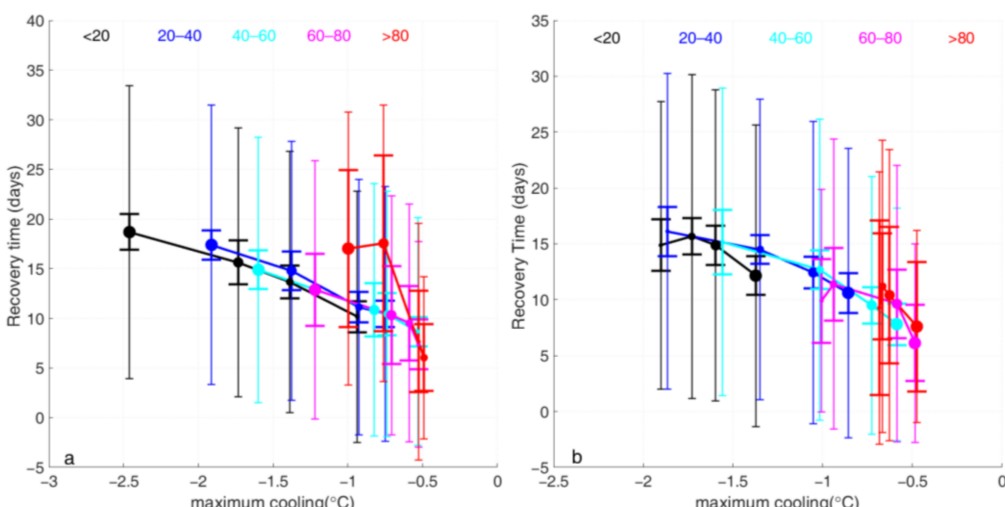

**Figure 7.** The average and standard deviation of maximum cooling and recovery time for each MLD bin for different (**a**) TC intensity and (**b**) translation speed (unit: m/s). The sizes of the circles in (**a**) represent the TD (smallest), TS (next to smallest), STS (next to largest), and TY (largest). The sizes of circles in (**b**) represent the translation speed of the TCs with <2.5 m/s (smallest), 2.5–5 m/s (next to smallest), 5–7.5 m/s (next to largest) and >7.5 m/s (largest). The thin and bold error bars represent the standard deviation of the individual value and the mean, respectively. The standard deviation of the mean is multiplied by 5 in the plot.

**Table 3.** The mean maximum cooling (°C) and recovery time (days, the number in brackets) for different MLD bins for different TC intensities. TD, TS, STS, and TY represent tropical depression (MSW under 33 kn), tropical storm (MSW between 34 kn and 47 kn), severe tropical storm (MSW between 48 kn and 63 kn), and typhoon (MSW above 64 kn), respectively.

| MLD Bin | TD | TS | STS | TY |
|---|---|---|---|---|
| 1 0–20 m | $-0.94 \pm 0.67$ $(10.2 \pm 12.7)$ | $-1.39 \pm 0.88$ $(13.7 \pm 13.2)$ | $-1.73 \pm 1.18$ $(15.6 \pm 13.5)$ | $-2.46 \pm 1.37$ $(18.7 \pm 14.7)$ |
| 2 20–40 m | $-0.75 \pm 0.61$ $(10.5 \pm 12.8)$ | $-0.93 \pm 0.70$ $(11.1 \pm 12.8)$ | $-1.38 \pm 0.90$ $(14.8 \pm 13.0)$ | $-1.91 \pm 1.23$ $(17.4 \pm 14.1)$ |
| 3 40–60 m | $-0.53 \pm 0.34$ $(8.7 \pm 11.5)$ | $-0.74 \pm 0.52$ $(10.4 \pm 12.4)$ | $-0.82 \pm 0.57$ $(10.8 \pm 12.7)$ | $-1.60 \pm 1.21$ $(14.9 \pm 13.4)$ |
| 4 60–80 m | $-0.53 \pm 0.40$ $(7.4 \pm 10.4)$ | $-0.59 \pm 0.34$ $(9.5 \pm 12.0)$ | $-0.71 \pm 0.50$ $(10.3 \pm 12.0)$ | $-1.22 \pm 1.07$ $(12.9 \pm 13.0)$ |
| 5 >80 m | $-0.53 \pm 0.32$ $(7.7 \pm 11.9)$ | $-0.49 \pm 0.29$ $(6.0 \pm 8.2)$ | $-0.76 \pm 0.34$ $(17.5 \pm 13.9)$ | $-1.00 \pm 0.61$ $(17.0 \pm 13.7)$ |

The results for dependence on TC translation speed are as expected: slower moving TCs drive larger maximum cooling and possess longer recovery times for all MLD bins. It is interesting to note that the maximum cooling for cases with the deepest MLD bin are nearly the same for all TCTS (Figure 7b). In addition, the recovery time of the slowest TCTS is shorter than that of the second slowest TC. However, this result may arise because the number of samples in the 60–80 m bin and >80 m bin of MLD is insufficient (Table 1). As shown in Table 4, increasing MLD has a more significant impact on maximum cooling for different TCTS situations. For example, for TCs translating at 0–2.5 m/s, the biggest difference in the maximum cooling is 1.21 °C between MLD bins 1 and 5, while it is only 0.89 °C between MLD bin 1 and 5 for TCs moving at 7.5 m/s.

**Table 4.** The average and standard deviation of maximum cooling (°C) and recovery time (the number in brackets, unit: day) for different MLD bins for different TC translation speeds.

| MLD Bin | 0–2.5 m/s | 2.5–5 m/s | 5–7.5 m/s | >7.5 m/s |
|---|---|---|---|---|
| 1 0–20 m | −1.90 ± 1.35 (14.9 ± 12.9) | −1.73 ± 1.25 (15.7 ± 14.5) | −1.60 ± 1.15 (14.9 ± 13.9) | −1.37 ± 1.05 (12.2 ± 13.5) |
| 2 20–40 m | −1.86 ± 1.40 (16.1 ± 14.1) | −1.35 ± 1.03 (14.5 ± 13.4) | −1.05 ± 0.83 (12.4 ± 13.5) | −0.86 ± 0.65 (10.6 ± 12.9) |
| 3 40–60 m | −1.56 ± 1.39 (15.2 ± 13.8) | −1.02 ± 0.86 (12.7 ± 13.5) | −0.73 ± 0.58 (9.5 ± 11.5) | −0.59 ± 0.39 (7.8 ± 10.4) |
| 4 60–80 m | −1.01 ± 1.09 (9.9 ± 10.0) | −0.94 ± 0.80 (11.4 ± 13.0) | −0.59 ± 0.41 (9.6 ± 12.4) | −0.49 ± 0.35 (6.1 ± 8.9) |
| 5 >80 m | −0.69 ± 0.38 (9.3 ± 12.2) | −0.67 ± 0.46 (11.2 ± 13.1) | −0.63 ± 0.45 (10.4 ± 13.0) | −0.48 ± 0.26 (7.6 ± 8.6) |

To investigate the spatial characteristics of the SST response to the passage of TCs in the northwestern Pacific, we calculated the spatial distribution of the TC induced maximum cooling, SST recovery time, TCI, TCTS, pre-passage MLD, and UTTG on a $1 \times 1$ degree grid (Figures 8 and 9). As shown in Figure 8a, the MLD is deep in the southeast and shallow in both the north and the South China Sea (SCS). The UTTG has the opposite pattern with the smallest values in the south and larger values to the north (Figure 8b). The maximum cooling shows a similar pattern to that of the MLD (Figure 9a), with small cooling in the southeast (less than 1 °C) and larger cooling in the SCS and to the north (Figure 9a). Both the MLD and UTTG have a moderate correlation with the maximum cooling, with correlation coefficients 0.44 and −0.33. In addition, the TCI has a moderate correlation with the maximum cooling as well, with a correlation coefficient of −0.36. However, all of the above correlations are significant at the 99% confidence level, based on a Student's t-test. Comparing the above three factors, the TCTS has little or no correlation with the maximum cooling ($r^2 = −0.003$), but this result may arise from other factors concealing its role as TCTS has a significant impact on maximum cooling for the same MLD as shown in Figure 7. The spatial pattern of the SST recovery time is opposite to that of the maximum cooling (Figure 9b), leading to a negative spatial correlation coefficient between them of −0.52, which is significant at the 99% confidence level. The results of the linear least squares fit show that the maximum cooling decreases ~0.1 °C with every 7 m increase in the MLD or every 0.04 °C/m decrease in the UTTG or every 5 m/s decrease in TCI or 2 m/s increase in TCTS.

Since there are relationships between MLD, UTTG, TCI, TCTS and the maximum cooling, and between the maximum cooling ($C_{max}$) and the recovery time ($T_r$), regression analyses were conducted to quantify these relationships. These analyses result in the following two relationships:

$$C_{max} = −0.902 + 0.015 \times MLD − 2.425 \times UTTG − 0.020 \times TCI + 0.061 \times TCTS \quad (3)$$

$$T_r = 5.245 − 6.167 \times C_{max} \quad (4)$$

The R-square of these two regression functions is 0.33 and 0.24, respectively. The bottom two panels in Figure 9 show the spatial distribution of the maximum cooling (Figure 9c) and the mean recovery time (Figure 9d) calculated from these regression formulas. Although the calculated maximum cooling and the recovery times are much smoother, they have a similar pattern as the observed results (compare Figure 9a to Figure 9c, and Figure 9b to Figure 9d). The spatial correlation coefficients are 0.57 and 0.52 for the maximum cooling and the recovery time, respectively, both of which are significant at the 99% confidence level. This indicates that the maximum cooling is primarily dependent on the MLD, UTTG, TCI, and TCTS, while the recovery time depends only on the maximum cooling. As a result, the magnitude and spatial pattern of the maximum cooling and the recovery time can be derived from the factors of the TCs and ocean thermal structure.

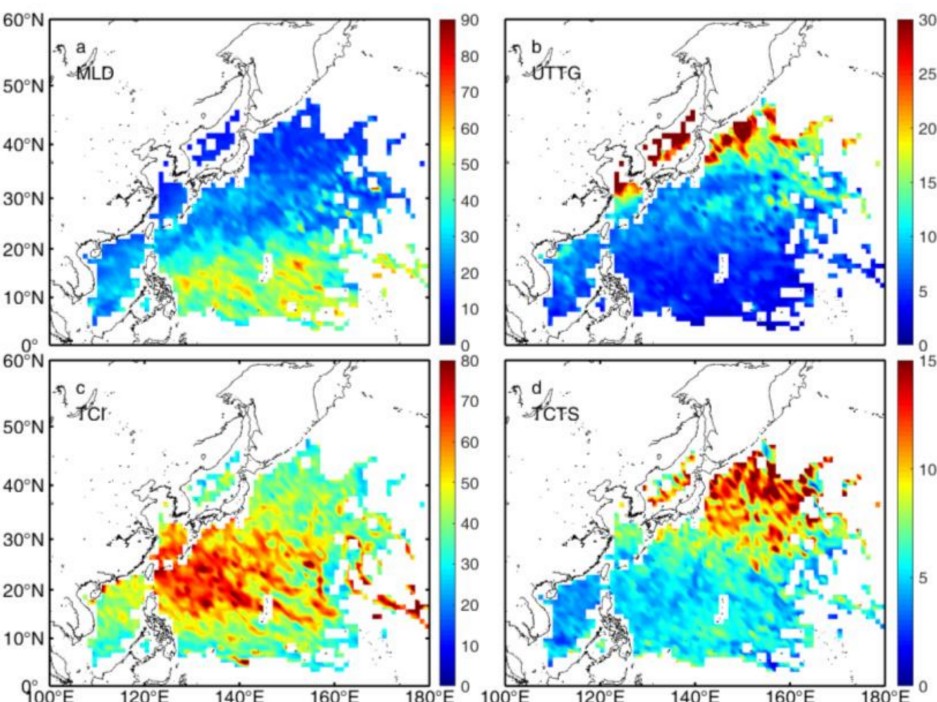

**Figure 8.** Spatial distribution of (**a**) MLD (unit: m), (**b**) UTTG (unit: $10^{-2}$ °C/m), (**c**) TC intensity (unit: m/s), (**d**) TC translation speeds (unit: m/s). All panels are observed/computed on a $1° \times 1°$ spatial grid.

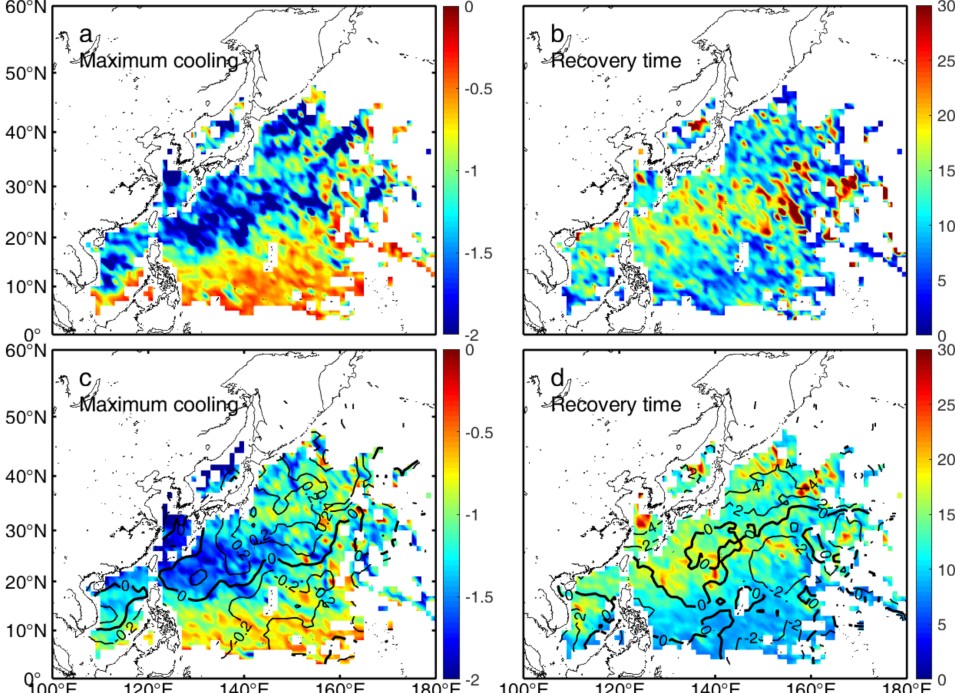

**Figure 9.** Spatial distribution of (**a**) observed maximum cooling (unit: °C), (**b**) observed recovery time (unit: day), and (**c**) regressed maximum cooling (unit: °C) calculated from the regression function on MLD, UTTG, TC intensity, and TC translation speeds and (**d**) regressed recovery time (unit: day) calculated from the regression function on maximum cooling. Contours in (**c**,**d**) represent the difference between observation and regressed values (regressed value minus observation) on a $3° \times 3°$ spatial grid. All panels are observed/computed on a $1° \times 1°$ spatial grid.

Comparing the spatial distribution of the regressed values to those of the observations, the error in maximum cooling is less than 0.2 °C in most areas with the maximum error exceeding 0.4 °C appearing in the northeastern area. The magnitude of regressed maximum cooling is larger (smaller) than the observations in the southern (northern) area (Figure 9c). For the recovery time (Figure 9d), the error is less than 4 days in most areas with the regressed recovery time being shorter in the southern area and longer in both the northern area and northeastern SCS. To further examine the results of the two regression functions, we compared the maximum cooling and recovery times calculated from the functions with the observation for each 0.5 °C bin of observed maximum cooling (Figure 10). As shown in panel 10a, the error increases almost linearly with the increase in the magnitude of the observed maximum cooling. The error is small in the −0.5 °C to −2.0 °C bins, with its peak value reaching 0.34 °C in the −2 °C bin. The ratio between the samples in these three bins and the total samples is more than 85%, indicating the regression result is reasonably good for most TC-induced SST cooling events. However, the error is large for strong TC-induced SST cooling events, which may result from the relatively small number of strong SST cooling events relative to the total number of events. For the recovery time, the error is small in −0.5 °C to −5°C bins, with a peak in the −5 °C bin of about 4.47 days. The samples in these bins account for 99% of the total samples, indicating the regression results are consistent with the observations. The above results show that the regression results are reasonably good for most TC-induced cooling events, but there still needs to be improvement for extreme SST cooling events.

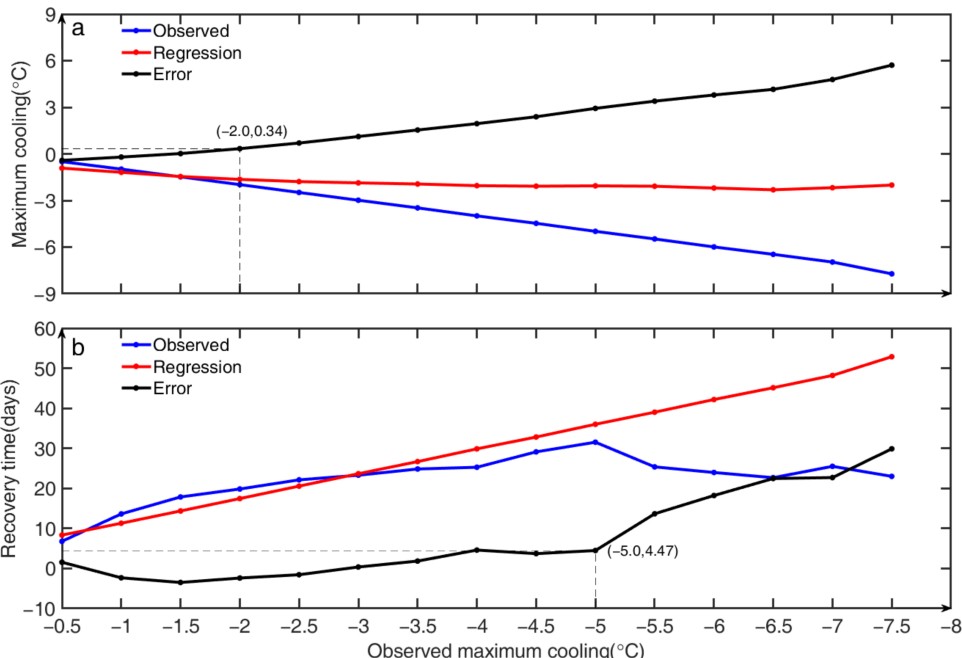

**Figure 10.** Comparison of maximum cooling (**a**) and recovery time (**b**) between observation and the regressed value in each 0.5 °C bin of observed maximum cooling.

## 4. Conclusions

The recovery of TC-induced SST cooling in NWP is investigated from both satellite and hydrographic observations. The results show that the passage of a TC induces SST cooling with an overall mean maximum cooling of −1.25 °C. In addition, it was found that most of this TC-induced cooling (~87%) will fade within 30 days. The recovery time was found to show a strong relationship with the maximum cooling, with the spatial correlation coefficient between them being −0.52.

The impacts of MLD and the UTTG on the SST response to the passage of a TC were also discussed. The MLD plays a significant role in setting both the maximum cooling

induced by the TCs and the following recovery time. The maximum cooling was found to increase nearly linearly with decreasing MLD such that a decrease of 7 m in the MLD leads to an increase in the maximum cooling of 0.1 °C. Further, longer recovery times are associated with shallower MLDs due to the larger maximum cooling. Since larger UTTGs are associated with a shallower MLD, the larger maximum cooling and longer recovery times are also associated with larger UTTG. The linear least squares result shows that the maximum cooling increases by 0.1 °C with a 0.04 °C/m increase in the UTTG, but changes in the UTTG have little or no significant effect on the recovery time. The combined effects of the MLD and TC's characteristics on the SST response to the TCs were also discussed in this paper. The results showed that the effect of MLD on SST cooling is directly related to the strength of the TC and that the maximum cooling associated with faster moving TCs is most affected by shallower MLD.

## 5. Discussion

The SST response to the passage of TCs is a complex process. As mentioned by Dare and McBride [30], both rapid and slow recovery processes can occur for all TCI and TCTS bins, which was also found to occur for all MLD and UTTG levels. This indicates that the entire oceanic thermal structure and varying TC characteristics introduce important factors affecting SST cooling. Further, the role that these factors play is complex and in addition to their individual effects, the interaction between them should be considered. However, due to the lack of Argo data during the passage of a TC, the MLD and UTTG used in this study had to be calculated from the monthly data, which introduces some uncertainty in the analyses. It is anticipated that a numerical model validated on an observational basis would be a good tool to use in a future study of the combined effect of these factors on the SST response to the passage of TCs. In addition, the recovery of TC-induced SST cooling can also meet complex air conditions such as sequential TCs and cold snap or ocean conditions such as mesoscale eddy and internal wave. These processes are not investigated in this study and would need further study. The detailed mechanisms of the recovery of TC-induced SST cooling are not investigated in the present study. The quantitative contributions from different dynamical/thermal dynamical processes under different conditions and hidden mechanisms should be further studied using an air–sea coupled model.

**Author Contributions:** Z.L. and Z.C. analyzed the data and prepared the manuscript; C.C., G.W. and H.H. provided important insights and suggestions on this research and re-edited the manuscript. All authors have read and agreed to the published version of the manuscript.

**Funding:** This paper was supported by the National Key Research and Development Program of China (2019YFC1510101), the Guangdong Provincial College Innovation, Team Project (2019KCXTF021), the First-class Discipline Plan of Guangdong Province (231419012, 231919030), the Oceanic Interdisciplinary Program of Shanghai Jiao Tong University (SL2020MS030).

**Institutional Review Board Statement:** Not applicable.

**Informed Consent Statement:** Not applicable.

**Data Availability Statement:** Publicly available datasets were analyzed in this study. These data can be found here: TC dataset is downloaded from https://www.ncei.noaa.gov/data/international-best-track-archive-for-climate-stewardship-ibtracs/v04r00/access (accessed on 1 November 2019). Satellite observed SST dataset is downloaded from www.remss.com (accessed on 12 April 2019). The monthly ocean temperature is downloaded from http://159.226.119.60/cheng/ (accessed on 14 September 2019). The Argo dataset is downloaded from ftp://ftp.argo.org.cn/pub/ARGO/global/ (accessed on 29 August 2020).

**Conflicts of Interest:** The researchers claim no conflict of interests.

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
