# Peer review of "Recovery of Tropical Cyclone Induced SST Cooling Observed by Satellite in the Northwestern Pacific Ocean"

_remotesensing, doi:10.3390/rs13183781_

Round 1

Reviewer 1 Report

Summary: The authors examine changes in satellite derived SST due to tropical cyclones, and how these changes depend on four variables. The initial results are quite interesting, and could be improved by including uncertainty analysis. These results might be impacted by seasonal change in SST. I strongly recommend that seasonal change in nearby SSTs by used to address this problem.  This is probably a relatively easy fix even if it causes all the statistics to be recalculated.  However, I have greater concerns about the fit of the concluding formulas to the data. I would like to see well these formulas fit the data, and if the differences from observations are due to a few outliers or there is a wide distribution. I would like to see this with a pdf or a scatter plot. Again, this seems like an easy change. Thus despite what I think are serous flaws to the paper I recommend that it be accepted with minor revisions. However, if the authors think that they cannot complete these revisions in time, I strongly recommend that they simply with draw it and resubmit it when they are ready.  I look forward to seeing the revised paper.

Major Comments:

  • Section 2.2: I am certain that for the longer time scale events that seasonal changes in the SST would modify these results. Has the seasonal change been accounted for? If so how? If not, how do you know it is not important?  While this might not be important in the 7 day examples, it seems essential to account for in the 50 day time series in Fig. 4.
  • Section 3.2: Are there cases where the MLD is limited by the depth of the ocean? If so, how are these accounted for?
  • Lines 302 to 320: While the correlations are statistically significant, the results appear to be physically insignificant. A correlation coefficient of 33% accounts for one nineth of the variability. Could the study be revised to better show the variability in the results and to illustrate the shape of the error distribution or the clustering of results?
  • Similarly, the results in equations (3) and (4) should be examined in a manner that shows how well this best fit actually fits the data. I suspect this analysis will impact the conclusions.

Minor Comments:

  • I suggest replacing TCI with ‘intensity,’ and replacing TCTS with ‘translational speed.’ While the acronyms help make the text shorter they are act to impair clarity. M:D is pretty common, so using that should not be a problem. UTTG might be replaced with ‘thermal gradient’ after explaining that it applies only to the mixed layer.
  • Line 78: it would be nice to see other agencies given credit as well as the CMA.
  • Line 89. What depth does the RSS SST product correspond to?  I believe it is foundation temperature rather than SST, which does impact the analysis.  This is important because it is not a skin temperature, but rather a temperature at the maximum depth where the diurnal cycle is negligible. This does not adversely impact the result, but does need to be explained. Arguably, this should help the interpretation.
  • Section 2.2: I am certain that for the longer time scale events that seasonal changes in the SST would modify these results.
  • Line 135: the word ‘data’ is plural. Please check the document for all occurrences of ‘data’ and fix this problem if needed. Or use dataset, which is singular.
  • Tables 1 and 2: Please remove the lines below the bin numbers. Please add uncertainty to each number (and say how it was determined).
  • 5-7: Please add error bars to each line.
  • Table 3: add uncertainty.
  • Lines 369 to 278: add a short explanation of TD, TS, STS and TY.
  • Line 289: This is too abrupt a change form intensity to translational speed. I suggest replacing ‘As expected,’ with ‘The results for dependence on translational speed are as expected:’.
  • Line 313: Why would we expect TCTS to impact cooling rate? It seems this would impact cooling, but not the rate of cooling, just as is found in this paper.

Author Response

Thanks for the positive evaluation and constructive comments.

Reviewer 2 Report

In this manuscript, the authors perform an extensive analysis of the cooling of sea temperature following a tropical cyclone in the Northwest Pacific. The data used are appropriate and have been treated statistically correctly and following an appropriate analysis methodology so that the results obtained are robust. It is particularly interesting that the authors calculate the cooling from the previous situation and not with respect to climatic values. 
In the development of their work, the authors show the complexity of the cooling process and the intervention of multiple factors. The different contributions and correlations between the different factors are correctly evaluated.
In short, this work is very complete and interesting and provides good knowledge about the field, so my recommendation is to publish it in its current state.

Only two minor points need to be corrected: 

  • Change "Day -1" to a less confusing indentifaction and remove white space. Maybe "D-1" or something similar.
  • Line 173: change "Figure 4a" to "FIgure 3a"

Author Response

Thanks for the positive comments.

Reviewer 3 Report

            In this paper, sea surface temperatures measured by satellite are used to study cooling induced by the passage of tropical cyclones in the Northwest Pacific Ocean. The authors considered how the magnitude and timing of the maximum cooling was influenced by the mixed layer depth (MLD) and upper thermocline temperature gradient (UTTG). They also considered the roles of tropical cyclone intensity and translation speed. They concluded that the maximum cooling magnitude negatively correlates with MLD, weakly positively correlates with UTTG and occurs ~1 day after tropical cyclone passage. Results from this study support prior studies that show that the tropical cyclone intensity is positively correlated with SST magnitude and translation speed. The Ling et al. study also examines the relationship between the oceanographic and storm factors and shows that a large MLD can modulate the magnitude and timing of maximum SST cooling.

            Overall, this paper is fundamentally sound. The methods the authors describe regarding using satellite data to determine sea surface temperature, using Argo float data and IAP data to construct temperature profiles, and controlling for antecedent sea surface temperatures (i.e. not just relying on a climatological mean) are good. Also, the conclusions that they draw based on the presented data are mostly substantiated (more on that point later). However, there are some flaws that need to be addressed.

            First, I found the discussion section to be too sparse. This is partially due to the fact that some discussion is sprinkled throughout the results section. But, mainly, there was very little discussion of the physical phenomena involved. For example, the authors concluded that UTTG had less of an influence on maximum SST cooling (both magnitude and timing) than MLD, but didn’t discuss why this might be so. Also, TC translation speed was examined, but again, there was no discussion about why slower storms might create larger maximum SST anomalies. Finally, I would have liked to see if the authors considered hysteresis in the SSTs (i.e. when a second storm passes through the same location before that location has recovered from the first storm, how would that affect SSTs?). It’s possible that there were too few cases of this to make robust conclusions, but this should be stated in the discussion.

            This brings me to my second point. There were places in the text (lines 270-271 and 292-293) where the authors mentioned that they had too few data points to draw a conclusion, but they didn’t state how many points they had. Also, at no point did they mention the number of tropical cyclones involved in this study. These are both important pieces of information that should be presented to the reader to help determine the robustness of the conclusions.

            In addition to these broad concerns, I have some specific concerns. First, figure 1 doesn’t seem to be a valid figure. The x-axis isn’t labeled, but the text mentions that it should be the Argo float number. There is no mention of what the x-axis means for the IAP data, so I’m inferring that it is the grid point closest to where the Argo data was collected. Unless I’m misunderstanding this, the Argo floats independently collect temperature profiles throughout the ocean. Therefore, the data for each float should be independent of the others. So, fitting a smoothing curve in figure 1 for the Argo data doesn’t make any sense. This is also true for the IAP data, which should all be independent grid points. Please either discard this figure or provide more information on the Argo floats and IAP data.

            Next, I would like to see more discussion about figure 4, in particular, why the magnitude of the SST cooling changes between figure 4A and 4B. Based on the description of figure 4B, it sounds like the authors shifted the temperature curves such that the maximum SST anomaly occurs at day 0. But I don’t understand how that would change the SST values (in the y-axis). Please elaborate on this.

            I also disagree with the author’s statement in the text (line 190) that longer recovery times are correlated with later maximum cooling since all of the curves in figure 4A show the maximum cooling on day 1. The sentence immediately after this one even conflicts with it.

            Further, in figures 5A and 6A, the cooling curves start at -0.5 °C one day prior to tropical cyclone passage. Please discuss why this should be the case. For example, was the tropical cyclone already creating a temperature anomaly before it reached this point? Was the SST below the mean climatological value? Or was some other mechanism at work?

            Another point that should be addressed in the methods section is why the bins for recovery time were chosen to be 1-5, 6-30, and >30 days.

            Finally, there are some minor errors that need to be addressed. They are:

  • Lines 34-39: These two sentences contain redundant information in that they both state that maximum SST cooling occurs on the right side of a tropical cyclone.
  • Figure 3: Please copy the x-axis values from 3B onto 3A since it isn’t immediately apparent that they’re the same (or reduce the space between 3A and 3B).
  • Table 3: Please define in the table caption what the storm strength acronyms mean
  • Figure 7: The size difference between the circles is too subtle. Either increase the sizes of some of the circles or use different shapes to convey this information
  • Line 173: There is a typo: the text should say “figure 3A” not 4A
  • Line 306: Please define “SCS” (I think it’s South China Sea?)
  • Line 345: Typo: This should refer to figure 9, not figure 10 (there isn’t a figure 10).

I believe this paper has some important conclusions that should be shared with the readers of this journal, but this article needs a lot more work.

Author Response

Thanks for the reviewer’s constructive comments on our paper.

Reviewer 4 Report

Review for "Recovery of tropical cyclone induced SST cooling observed by satellite in the northwestern Pacific Ocean"

Major comments:

This manuscript tends to reveal the combined effects of the oceanic thermal structure and properties of tropical cyclone (TC) on SST cooling. The roles of mixed layer depth (MLD), upper ocean thermal gradient below the MLD, intensity of TC, and its translation speed on the SST response are quantified.

This work provides useful information on the characteristics of SST variation by TC and on the portion of determining factors. It is meaningful that this study suggests a linear relation between the SST cooling and the determining factors. Also, the authors found that the linear relation between the SST cooling and recovery time. While the subject is interesting and meaningful to the studies of TC-ocean interactions, Introduction and discussion sections can be improved. 

Minor comments:

In Lines 37-38: It is redundant with Lines 34-35. Delete this sentence.

In Lines 38: Specify 'the opposite phenomenon'. Does it indicate surface warming? or left side cooling?

In Lines 56-57: This sentence is a main sentence of this paragraph. It is better to move this to the beginning of the paragraph

In Lines 67-70: Please refer related references about this topic.

e.g.,

Park, J. J., Kwon, Y. O., & Price, J. F. (2011). Argo array observation of ocean heat content changes induced by tropical cyclones in the north Pacific. Journal of Geophysical Research: Oceans116(C12).

Fu, H., Wang, X., Chu, P. C., Zhang, X., Han, G., & Li, W. (2014). Tropical cyclone footprint in the ocean mixed layer observed by A rgo in the N orthwest P acific. Journal of Geophysical Research: Oceans, 119(11), 8078-8092.

Lin, I. I., Black, P., Price, J. F., Yang, C. Y., Chen, S. S., Lien, C. C., ... & D'Asaro, E. A. (2013). An ocean coupling potential intensity index for tropical cyclones. Geophysical Research Letters40(9), 1878-1882.

In Line 74: Specify the period of the best track data and show the number of TCs analyzed in this study.

In Line 98: This should be moved to the method section.

In Line 108: There is a possibility that the climatological SST contains the effect of TC-induced SST cooling because the occurrence of TC has a seasonal signal. With the consideration of the effect, please justify the method.

In Line 113: STT->SST.

If you want to use this abbreviation, please define Delta SST_{TC}

In Line 139: that of -> those of

In Line 144: please refer equations 1 and 2

In Figure 1 & 4: Please insert a label of X-axis

In Figure 7: Please insert a and b in the panels

In Line 345: Figure 10 -> Figure 9

In Discussion section, more discussion about the meaning of this study from the related papers are needed.

Author Response

Thanks for the positive comments.

Round 2

Reviewer 1 Report

Summary: The authors responded well to the prior review.  Thank you for the thoughtful and kind response.  The revised paper is remarkably better than the original draft and is close to the expectations for publication.

Major Comments:

1) The error bars do a great job a showing the spread, which greatly improves the paper. However, they probably misleadingly suggest that there is low confidence in means with a large number of samples. I suggest also adding error bars that show the uncertainty in the mean.  I realize that this might not be practical if the distributions are difficult to model. If that is the case, I suggest simply determining the uncertainty for the mean of a Gaussian distribution and stating why this is not ideal. Although I hope that the distribution can be reasonably fit, and that the fitting parameters can be used to determine the uncertainty in the mean.

Minor Comments:

2) Lines 77 and 78: Please change “is used in the present study according to the previous studies” to “is used in the present study as justified by previous studies…”

Author Response

Thanks for the positive comments.

Reviewer 3 Report

The improvements made by the authors have greatly increased the clarity of this manuscript. The figures are now easier to understand and, in some cases, convey more information than the previous versions (e.g. adding standard deviation error bars). The expanded discussion also helps the reader understand the modeling results and how they compare with SST observations. I have no further reservations or recommendations.

Author Response

Thanks for the positive comments.